# Oxidative Stress-Induced Sirtuin1 Downregulation Correlates to HIF-1α, GLUT-1, and VEGF-A Upregulation in Th1 Autoimmune Hashimoto’s Thyroiditis

**DOI:** 10.3390/ijms22083806

**Published:** 2021-04-07

**Authors:** Michaël Hepp, Alexis Werion, Axel De Greef, Christine de Ville de Goyet, Marc de Bournonville, Catherine Behets, Benoit Lengelé, Chantal Daumerie, Michel Mourad, Marian Ludgate, Marie-Christine Many, Virginie Joris, Julie Craps

**Affiliations:** 1Pole of Morphology, Faculty of Medicine, Institute of Experimental and Clinical Research, Université Catholique de Louvain, B-1200 Brussels, Belgium; michael.hepp@student.uclouvain.be (M.H.); alexis.werion@student.uclouvain.be (A.W.); axel.degreef@student.uclouvain.be (A.D.G.); christine.deville@uclouvain.be (C.d.V.d.G.); marc.debournonville@uclouvain.be (M.d.B.); catherine.behets@uclouvain.be (C.B.); benoit.lengele@uclouvain.be (B.L.); julie.craps@uclouvain.be (J.C.); 2Endocrinology Department, Faculty of Medicine, Université Catholique de Louvain, B-1200 Brussels, Belgium; chantal.daumerie@uclouvain.be; 3Surgery and Abdominal Transplantation-Utragendo Department, Faculty of Medicine, Université Catholique de Louvain, B-1200 Brussels, Belgium; michel.mourad@uclouvain.be; 4Thyroid Research Group, Division of Infection & Immunity, Cardiff University School of Medicine, Heath Park, Cardiff CF14 4XN, UK; Ludgate@cardiff.ac.uk; 5Pole of Pharmacology and Therapeutics, Faculty of Medicine, Institute of Experimental and Clinical Research, Université Catholique de Louvain, B-1200 Brussels, Belgium; virginie.joris@uclouvain.be

**Keywords:** Hashimoto’s thyroiditis, oxidative stress, NOX4, Sirtuin1, HIF-1α

## Abstract

In Hashimoto’s thyroiditis (HT), oxidative stress (OS) is driven by Th1 cytokines’ response interfering with the normal function of thyrocytes. OS results from an imbalance between an excessive production of reactive oxygen species (ROS) and a lowering of antioxidant production. Moreover, OS has been shown to inhibit Sirtuin 1 (SIRT1), which is able to prevent hypoxia-inducible factor (HIF)-1α stabilization. The aims of this study were to determine the involvement of NADPH-oxidases (NOX), SIRT1, and HIF-1α in HT pathophysiology as well as the status of antioxidant proteins such as peroxiredoxin 1 (PRDX1), catalase, and superoxide dismutase 1 (SOD1). The protein expressions of NOX2, NOX4, antioxidant enzymes, SIRT1, and HIF-1α, as well as glucose transporter-1 (GLUT-1) and vascular endothelial growth factor A (VEGF-A), were analyzed by Western blot in primary cultures of human thyrocytes that were or were not incubated with Th1 cytokines. The same proteins were also analyzed by immunohistochemistry in thyroid samples from control and HT patients. In human thyrocytes incubated with Th1 cytokines, NOX4 expression was increased whereas antioxidants, such as PRDX1, catalase, and SOD1, were reduced. Th1 cytokines also induced a significant decrease of SIRT1 protein expression associated with an upregulation of HIF-1α, GLUT-1, and VEGF-A proteins. With the exception of PRDX1 and SOD1, similar results were obtained in HT thyroids. OS due to an increase of ROS produced by NOX4 and a loss of antioxidant defenses (PRDX1, catalase, SOD1) correlates to a reduction of SIRT1 and an upregulation of HIF 1α, GLUT-1, and VEGF-A. Our study placed SIRT1 as a key regulator of OS and we, therefore, believe it could be considered as a potential therapeutic target in HT.

## 1. Introduction

Hashimoto’s thyroiditis (HT) and Graves disease are autoimmune thyroid diseases, which comprise around 30% of all autoimmune conditions. HT is the principal cause of hypothyroidism and goiter in a population where the diet supplies a sufficient iodine level [1]. In 90% of HT patients, high serum levels of thyroid peroxidase antibodies are detectable, whereas increased concentrations of serum thyroglobulin antibodies are present in only 20% to 50% of such cases [2]. 

The known autoimmune reaction in HT implicates antibody-producing B lymphocytes as well as T cells. CD4+ T lymphocytes play a key role in the pathogenesis of HT leading to T helper type 1 (Th1) stimulation. Th1 cytokines, such as interferon (IFN)γ, interleukin (IL)-1α, IL-1β, Il-2, tumor necrosis factor (TNF)-α, and TNF-β, trigger thyrocyte destruction by CD8+ cytotoxic cells, which ultimately leads to gland atrophy and hypothyroidism [3].

Thyrocyte destruction in HT is associated with oxidative stress [4,5,6]. As in other diseases, oxidative stress (OS) is due to an increased production of reactive oxygen species (ROS) such as H_2_O_2_, superoxide anion, and due to a lowering of the numerous antioxidant defenses, such as peroxiredoxins (PRDX) and catalase, which detoxify H_2_O_2_, and superoxide dismutase 1 (SOD1), transforming superoxide in H_2_O_2_ [7].

The source of ROS in HT is still debated. In HT samples, as well as in human thyroid cells treated with Th1 cytokines, Marique et al. (2014) demonstrated a downregulation of Caveolin 1 (Cav-1), inducing the disruption of the thyroxisome, a multiprotein complex containing thyroid peroxidase (TPO) and dual oxidase (DUOX), which must be located at the apical membrane to be effective in terms of efficient hormone synthesis [8]. This results in a mislocalization of DUOX and the subsequent production of H_2_O_2_ inside the cytoplasm rather than at the apical pole facing the colloid. Moreover, Werion et al. (2016) highlighted that a decrease in PPARγ, a well-known regulator of Cav-1 and also of the antioxidant catalase, is a key event in HT pathogenesis [9].

Colin et al. (2014) proposed that ROS production induced by Th1 cytokines in thyroid cell lines results from NOX2 upregulation and, thus, superoxide overproduction [10]. They did not observe any modification of NOX4 expression in the PCCL3 cell lines and in the thyroid samples of two HT patients.

NOX4, known to generate H_2_O_2_ (90%) and superoxide (10%), has been detected in human thyrocytes [11]. Furthermore, this enzyme is overexpressed in papillary thyroid cancers (PTC) [11,12] and chronic ROS exposure has been proposed to explain the predisposition of HT patients to develop thyroid carcinoma [13]. NOX4-related ROS production is also recognized as an essential component in the metabolic adaptation of PTC cells via HIF-1α increase and consecutive glucose transporter 1 (GLUT-1) upregulation [14]. In thyroid cells, HIF-1α is also known to be implicated in VEGF-A upregulation and angiogenesis [15,16] with an increase of VEGF-A being a hallmark of thyroid carcinomas [17].

A large number of physiological and pathological phenomena linked to OS implicate Sirtuin (SIRT)1, a Nicotinamide adenine dinucleotide (NAD)^+^-dependent deacetylase. It has been demonstrated that SIRT1 controls ROS levels and vice versa. Among ROS, H_2_O_2_ plays important roles by downregulating SIRT1 enzymatic activity through a reduction of intracellular NAD^+^ levels [18]. H_2_O_2_ also decreases SIRT1 protein expression and induces a nucleocytoplasmic shuttling of SIRT1, leading to proteasome degradation [18,19]. In addition, the intracellular glutathione/glutathione disulfide (GSH/GSSG) ratio, and thus the intracellular redox status, may influence SIRT1 deacetylation function [20]. 

Currently, there are very few reports on the role of SIRT1 in HT and its possible interaction with the OS characteristic of this disease. Al-Khaldi and Sultan (2019) [21] showed a decrease of SIRT1 mRNA expression in serum of HT patients and Roehlen et al. (2018) [22] suggested the involvement of the FOXO3a-SIRT1 pathway in the positive effect of Vitamin D on HT.

This study aimed to verify whether SIRT1 is involved in the modulation of oxidative stress and angiogenesis occurring in Hashimoto’s disease. We performed in vitro experiment on human thyroid cells in the presence or absence of Th1 cytokines and on ex vivo controls as well as on HT human thyroid samples. We analyzed (1) the expression of NOX2 and NOX4 as sources of ROS, (2) the expression of antioxidants PRDX1, catalase, and SOD1, (3) the expression of SIRT1 protein in the context of the HT oxidative stress, and (4) the expression of HIF-1α and its targets, GLUT-1 and VEGF-A.

## 2. Results

### 2.1. Th1 Cytokines Increase ROS Production and NOX4 Protein Expression in Human Thyroid Cells, without Changing NOX2 Protein Expression

ROS were detected by the 2′,7′-dichlorodihydrofluorescein diacetate (CM-H_2_DCFDA) method in human thyrocytes treated with IL-1α and IFNγ to mimic Hashimoto’s thyroiditis (HT) and compared to non-treated cells. ROS were lightly marked by green fluorescence in the cytoplasm of untreated control cells (Figure 1A). In Th1 cytokine-treated thyrocytes, ROS expression was highly increased (Figure 1B). A significant rise of ROS was confirmed by the quantification of fluorescence normalized to the number of nuclei (Figure 1C). ROS excess could be due to an increase of NOX4 protein known to generate H_2_O_2_ by 90% and superoxide by 10%. Th1 cytokines induced a significant increase of NOX4 protein expression measured by Western blotting (Figure 1D). However, no change was observed for NOX2 protein expression (Figure 1E). 

### 2.2. NOX4 Protein Expression Is Increased in HT Thyroid Samples without Any Change of NOX2 Protein Expression

Thyroid samples from both control and HT patients displayed considerable heterogeneity in expression of NOX2 and NOX4 quantified by Western blot. We noticed, however, a significant increase of NOX4 (Figure 2A), whereas NOX2 was unmodified as compared to controls (Figure 2B). 

As previously shown by Gérard et al. (2002), a normal thyroid gland displays considerable morphological and functional heterogeneity among follicles, with hypofunctional follicles (†) lined by flat thyrocytes and active follicles (*) recognized by their small size and their thickened epithelial layer [23]. In HT thyroids, the follicular heterogeneity was even more accentuated with hypofunctional follicles (†) and altered active-like follicles ($), the latter presenting the same morphological aspects as those that were described by Marique et al. (2014) as having lost their capacity for normal hormonal synthesis and showing a high expression of 4-Hydroxynonenal (HNE), a marker of OS [8]. We confirmed in our samples that active-like follicles lost their capacity for T4 synthesis and strongly expressed HNE (Appendix A). 

In paranodular tissues of multinodular goiter considered as controls, NOX4 protein expression was low in hypofunctional follicles while the active follicles revealed greater staining (Figure 2C). In HT samples, NOX4 staining was increased in the active-like follicles (Figure 2D,E). NOX2 immunostaining was observed in active follicles from controls (Figure 2F) and in active-like follicles from HT glands (Figure 2G) but it was mainly increased in the numerous inflammatory cells infiltrating the thyroids of HT patients (Figure 2H).

### 2.3. The Expression of PRDX1 and Catalase Is Decreased by Th1 Cytokines in Human Thyroid Cells and the Expression of Catalase Is Lowered in HT Thyroid Samples

PRDX1 and catalase protein expression quantified by Western blotting were significantly decreased in primary cultures of human thyrocytes treated with Th1 cytokines (Figure 3A,B). In thyroids of HT patients, PRDX1 protein expression was unchanged (Figure 3C) but catalase expression was significantly decreased as compared to controls (Figure 3D).

In control thyroids, PRDX1 (Figure 3E) and catalase (Figure 3H) were detected in both hypofunctional and active follicles although the latter showed a higher degree of immunostaining. In HT non-inflammatory zone, the expression of PRDX1 (Figure 3F) and catalase (Figure 3I) was reduced compared to controls. In HT inflammatory zones, the expression of PRDX1 (Figure 3G) and catalase (Figure 3J) was high in active-like follicles in which the OS is maximal. 

### 2.4. The Expression of SOD1 Is Decreased by Th1 Cytokines in Human Thyroid Cells and Is Unchanged in HT Thyroid Samples

Western blotting quantification revealed that SOD1 protein expression was significantly diminished by Th1 cytokines’ treatment (Figure 4A) but SOD1 was not modified in HT thyroid samples compared to controls (Figure 4B). In control, as well as in HT thyroids, SOD1 protein expression was detected by immunohistochemistry (IHC) in hypofunctional follicles and in active or active-like follicles in which the staining was intense (Figure 4C–E). 

### 2.5. In Th1 Cytokines-Treated Human Thyroid Cells, as Well as in HT Thyroid Samples, the Expression of SIRT1 Is Decreased While the Expression of HIF-1α Is Increased

A significant decrease of SIRT1 protein expression was observed in primary cell cultures of human thyrocytes treated with Th1 cytokines (Figure 5A). Thyrocytes treated with IL-1α and IFNγ showed a highly increased HIF-1α protein expression compared to non-treated cells (Figure 5B). In whole thyroid samples of HT patients, Western blotting quantification revealed reduced SIRT1 protein expression (Figure 5C) and upregulated HIF-1α protein expression (Figure 5D) compared to controls. 

Immunostaining showed that SIRT1 protein expression was high in active follicles of control glands (Figure 5E) and slightly decreased in active-like follicles from HT thyroids (Figure 5F,G). The staining of HIF-1α was higher in HT thyroids (Figure 5I,J) than in controls (Figure 5H).

### 2.6. The Expression of GLUT-1 and VEGF-A Is Increased by Th1 Cytokines in Primary Cultures of Human Thyrocytes and in HT Thyroid Samples

Thyrocytes treated with Th1 cytokines had higher GLUT-1 and VEGF-A protein expressions than non-treated thyrocytes (Figure 6A,B). Western blotting quantification revealed that in HT patients, GLUT-1 and VEGF-A protein expression was also significantly increased compared to controls (Figure 6C,D).

Compared to controls (Figure 6E), a higher immunohistochemical staining of GLUT-1 was observed in HT thyroids (Figure 6F,G). Similar results were observed for VEGF-A (Figure 6H–J). Once again, active and active-like follicles showed more intense cytoplasmic staining than hypofunctional follicles.

## 3. Discussion

Our results demonstrated that, in primary cultures of human thyroid cells, Th1 cytokines induce an increase in NOX4 known to generate H_2_O_2_ and superoxide as well as a decrease in antioxidant proteins, PRDX1, catalase, and SOD1, responsible for OS. This is associated with a downregulation of SIRT1 protein expression and an upregulation of HIF-1α, GLUT-1, and VEGF-A.

In Hashimoto’s thyroid samples, the proteins studied followed the same trends. However, follicular heterogeneity was apparent, even in normal thyroids, including hypofunctional follicles and active follicles, with the latter being characterized by their small size and their thick thyrocytes [23]. The intraglandular heterogeneity was even more accentuated in HT with hypofunctional follicles and active-like follicles that, as described by Marique et al. [8], may be considered as altered follicles having lost their function and being the targets of OS, as shown by HNE immunostaining. 

Together, with the large follicular heterogeneity, there were also regional differences in the inflammation within the glands of HT patients. Of course, this is a limitation of our study in which whole glands were homogenized to perform Western blots. However, except for PRDX1 and SOD1, we observed significant differences between controls and HT patients. 

Our study shows that NOX4 was increased in human thyroid cells treated with Th1 cytokines, as well as in HT thyroid samples, while NOX2 expression was not modified. This is in contrast to the results reported by Colin et al. (2014) [10] and this could be explained by the fact that we used primary cultures of human thyroid cells rather than PCCL3 cell lines. Colin et al. (2014) [10] analyzed the NOX2 protein level by immunohistochemistry on two HT patients without making the distinction between the different types of follicles. As illustrated in this paper, these proteins revealed considerable heterogeneity in their expression. NOX4 was mainly increased in active-like follicles, whereas NOX2 was strongly expressed in inflammatory cells. NOX4 expression was shown to be regulated in vascular cells by SIRT1 [24,25], which also reduced OS by increasing antioxidant enzymes such as SOD1 and catalase [26,27,28].

Our results showed a decrease of SIRT1 in Th1-treated human thyroid cells and in Hashimoto’s thyroids, suggesting that a significant interaction between ROS and SIRT1 exists in HT. H_2_O_2_ could hold a key role in this interaction between OS and SIRT1. Indeed, this ROS could be produced in Th1 cytokines-treated human thyrocytes and in HT by a decrease of Cav-1 [8] or PPARγ [9] and also by a NOX4 overexpression [11]. However, H_2_O_2_ is most likely not the only ROS implicated and ROS are not the only explanation for a SIRT1 decrease. Several authors showed that post-translational modifications, inflammatory cytokines, and miRNA could also explain SIRT1 protein decrease in an OS environment [18,29]. 

SIRT1 is also known to interact with HIF-1α [30] as it inhibits its cellular accumulation via deacetylation, leading to an impairment of the resulting signaling pathways [31,32,33]. In consequence, hypoxia-induced suppression of SIRT1 may be responsible for an improvement of HIF-1α signaling [31]. Although the exact mechanism remained to be elucidated, our results showed a correlation between SIRT1 downregulation and HIF-1α upregulation in Th1 cytokines-treated human thyroid cells and in HT patients. 

In normoxic conditions, HIF-1α has a cytoplasmic localization and is rapidly degraded by the proteasomes before being able to act in its role as transcription factor in the nucleus. Under hypoxic conditions, HIF-1α is stabilized and translocated to the nucleus where HIF-1α/HIF-1β binding occurs and gene transcription can be initiated [34]. Hypoxia can stabilize HIF-1α but nuclear factor (NF)-κB factors and cytokines can also lead to a HIF‑1α signaling [35,36]. Consequently, HIF-1α appears to be overexpressed in many malignant tumor types, whereas in normal tissues, lower protein levels were observed [34,37,38,39]. 

Furthermore, it is known that HIF-1α acts as a GLUT-1 transcription factor [14,40]. Indeed, aerobic glycolysis is another well-known and major process observed in cancer metabolism [14,41] but our work also highlighted a similar upregulation in HT tissues. The upregulation of VEGF-A that we observed in Th1 cytokines-treated human thyrocytes and in HT thyroid samples could also be induced by HIF-1α. Indeed, VEGF is another well-known target of HIF-1α in thyroid cells [15,16] and it is known to be increased in HT [42] and in thyroid cancers [17]. 

To date, it is increasingly accepted that thyroid cancers develop in patients with autoimmune thyroid disease [43,44,45] and preferentially in patients with HT [46]. Our results support this observation by showing that, as in papillary thyroid carcinoma (PTC), NOX4, HIF-1α, GLUT-1, and VEGF-A are increased in Th1 cytokines-treated human thyroid cells and in Hashimoto’s thyroid samples. The similar expression pattern between HT and PTC may suggest that the increased metabolism associated with a disturbed redox balance in HT thyrocytes could potentially be a first step toward tumorigenesis. 

In conclusion, as illustrated in Figure 7, we propose a pathogenic pathway for HT development centered on ROS overproduction and SIRT1 downregulation. Intracellular increase of ROS could result from an upregulation of NOX4 induced by Th1 cytokines. Our results also suggest that the element linking the OS and the HIF-1α/GLUT-1 and HIF-1α/VEGF-A pathways may involve SIRT1. An OS-induced downexpression of SIRT1 may lead to increased HIF-1α activity followed by overexpression of GLUT-1 and VEGF-A. Moreover, the diminished SIRT1 expression induced a reduction of antioxidant defenses due to PRDX1, catalase, and SOD1, further aggravating OS. Interestingly, in accordance with our observations, it was shown that SIRT1 regulates NOX4 expression [24,25], which is a protein considered essential for OS in thyrocytes. Finally, SIRT1, as an antioxidative stress molecule, could be considered as an optimal therapeutic target for HT.

## 4. Materials and Methods

### 4.1. Patients 

Thyroid tissues from HT patients (*n* = at least 5, Table 1) were surgically obtained for Western blotting (WB) and immunohistochemistry (IHC). Paranodular tissue from patients with multinodular goiter was used as control samples (*n* = at least 5) and used to perform primary cell cultures. All thyroid samples were obtained after patients gave their informed consent (4 October 2017/466). Samples were obtained from the Biolibrary of the Cliniques universitaires Saint-Luc, referenced as BB190044, member of the Biothèque Wallonie Bruxelles (BWB) and of biobanking and biomolecular resources research infrastructure in Belgium (BBMRI.be). 

### 4.2. Primary Cell Cultures

As previously described [16], thyrocytes were isolated from paranodular tissue from patients operated on for multinodular goiter. Briefly, thyroid samples were cut into pieces and incubated several times in collagenase type II (Worthington, Lakewood, CA, USA) solution until entirely digested. Isolated cells were filtered and cultured in modified Earle’s medium (MEM) (ThermoFischer Scientific, Waltham, MA, USA) to which 5% newborn calf serum (ThermoFischer Scientific, Waltham, MA, USA), 1% penicillin–streptomycin (ThermoFischer Scientific, Waltham, MA, USA), 1% amphotericin B (ThermoFischer Scientific, Waltham, MA, USA), and 1 mU/mL TSH (Sigma-Aldrich^®^, St. Louis, MO, USA) at 37 °C, 5% CO_2_ were added. After five days, cells were starved for one day in medium containing 0.5% newborn calf serum and then incubated for one additional day with recombinant human IL-1α (2 ng/mL, PeproTech, Minneapolis, MN, USA) and recombinant human IFNγ (10 ng/mL, PeproTech, Minneapolis, MN, USA) diluted in MEM containing 0.5% newborn calf serum. Cells were then harvested for further analyses. Primary cell cultures were realized at least in duplicate and repeated at least three times. 

### 4.3. Immunohistochemistry

Immunostaining was performed as previously described [16]. Briefly, human thyroid samples were fixed in 4% paraformaldehyde and then embedded in paraffin. After rehydration, paraffin sections underwent antigen retrieval by heating in citrate buffer (not done for HIF-1α staining) and endogenous peroxidase activity was inhibited with 1% H_2_O_2_ solution for 20 min. Paraffin sections were blocked with a goat serum (1/50, Sigma-Aldrich^®^, St. Louis, MO, USA) and Bovine serum albumin (BSA) (Millipore, Merck, Darmstadt, Germany) 5% solution for 30 min at room temperature (RT). Sections were then incubated with primary antibody. Incubation time and dilution varied depending on the primary antibody used, as described in Table 2. A secondary antibody, Envision (Dako, Carpinteria, CA, USA), coupled to peroxidase and specific for the primary antibody (Table 2) was applied to the slides for 1 h at RT. Peroxidase activity was revealed with 3-3′ diaminobenzidine tetrahydrochloride (DAB) (Vector, Burlingame, CA, USA). Sections were then counterstained with Mayer’s hematoxylin and fixed in Entellan mounting medium (Millipore, Merck, Darmstadt, Germany).

### 4.4. Western Blotting

The thyroid samples or cultured cells were homogenized in radioimmunoprecipitation assay (RIPA) buffer containing a protease inhibitor cocktail and PhoSTOP (Roche, Basel, Switzerland). Protein concentration was determined using a bicinchoninic assay protein assay kit (ThermoFischer Scientific, Waltham, MA, USA). For Western blotting [47], 10 µg of proteins were loaded per well on a SDS-PAGE gel and migration was performed in migration buffer (Tris-EDTA, SDS, glycine) under 120 V for 90 min. Proteins were transferred onto nitrocellulose blotting membranes (GE Healthcare, Chicago, IL, USA) with a constant amperage of 350 mA for 90 min in transfer buffer (Tris-base, glycine, SDS, methanol). Membranes were blocked for 1 h at RT with TBS (Tris buffer Saline) containing 0.1% Tween and 5% BSA. Primary antibodies (Table 2) were then incubated on membranes overnight at 4 °C for immune detection of targeted proteins. Membranes were washed in TBS-Tween and incubated for 1 h at RT with the peroxidase-conjugated secondary antibody. Finally, membranes were developed on CL-X Posure^®^ films (ThermoFischer Scientific, Waltham, MA, USA) using ECL^TM^ Western Blotting Detection Reagents (GE Healthcare, Chicago, IL, USA). Protein levels were normalized to β-actin for primary cultures of thyroid cells. The high number of inflammatory cells and connective tissue in human samples forced us to use GAPDH as normalizer for these blots as β-actin is more highly expressed in inflammatory cells in the fibrotic state [48,49]. Films were scanned and quantified by densitometry using ImageJ software (Version 1.52A) [50].

### 4.5. Quantification of ROS Levels Using CM-H2DCFDA

Levels of ROS in primary cultures of human thyrocytes, treated or not with Th1 cytokines, were assessed using CM-H_2_DCFDA technology. Briefly, human primary thyrocytes were cultured on glass coverslips and treated during 24 h with Th1 cytokines (IL-1α and IFNγ) after six days of culture including 24 h of starvation. At the day of analysis, cells were washed with Hank’s Balanced Salt Solution (HBSS, ThermoFisher Scientific, Waltham, MA, USA) and incubated in a 10-µM solution of 6‑chloromethyl‑2′,7′‑dichlorodihydrofluorescein diacetate (CM-H_2_DCFDA, ThermoFisher Scientific, Waltham, MA, USA) at 37 °C in the dark for 30 min. Fluorescence was visualized with a fluorescence microscope by excitation around 488 nm. Images were quantified using the ImageJ program (Version 1.52a) [50].

### 4.6. Statistical Analysis

All experiments were reproduced at least three times. All values are expressed as means ± SEM. Shapiro–Wilk normality test was performed, and statistically significant differences were determined using a student t-test or by using a Mann–Whitney test for non-parametric data. *p* < 0.05 was considered statistically significant.

## Figures and Tables

**Figure 1 ijms-22-03806-f001:**
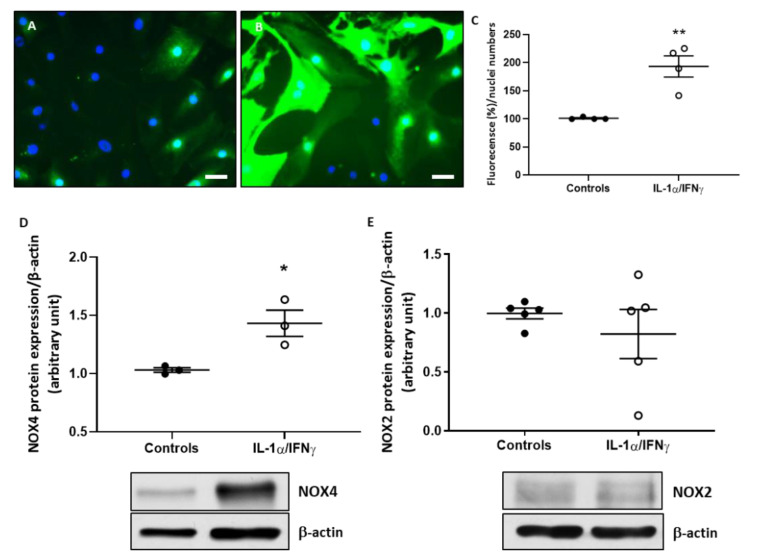
ROS production and NOX4 protein expression increased in primary cultures of human thyrocytes treated with Th1 cytokines, mimicking Hashimoto’s thyroiditis, while NOX2 protein expression did not change. Human thyrocytes were treated for 24 h with Il-1α and IFNγ. Cells were immersed in a 1-µM solution of CM-H2DCFDA for 30 min. Nuclei were stained by Hoescht. Images shown are representative of each condition. The picture on the left shows non-treated control thyrocytes (**A**) and Th1 cytokines-treated thyrocytes are displayed on the right (**B**). Scale bar = 10 µm. ROS levels were increased in thyrocytes treated with Th1 cytokines compared to controls. Quantification of ROS expression was analyzed using ImageJ software (**C**). Fluorescence was quantified and normalized to the number of nuclei. Results are expressed as means ± standard error of the mean (SEM) from four experiments (*n* = 4). ** *p* < 0.01 compared to control. NOX4 (**D**) and NOX2 (**E**) protein expressions were quantified by Western blot technique in human thyrocytes treated or not with Th1 cytokines. Th1 cytokines induced a significant increase of NOX4 expression while NOX2 did not change. Densitometric values were normalized against the β-actin level. Results are expressed as means ± SEM from three (NOX4) and five (NOX2) experiments (*n* = 3–5) at least in duplicate. * *p* < 0.05 compared to control. A representative blot is shown.

**Figure 2 ijms-22-03806-f002:**
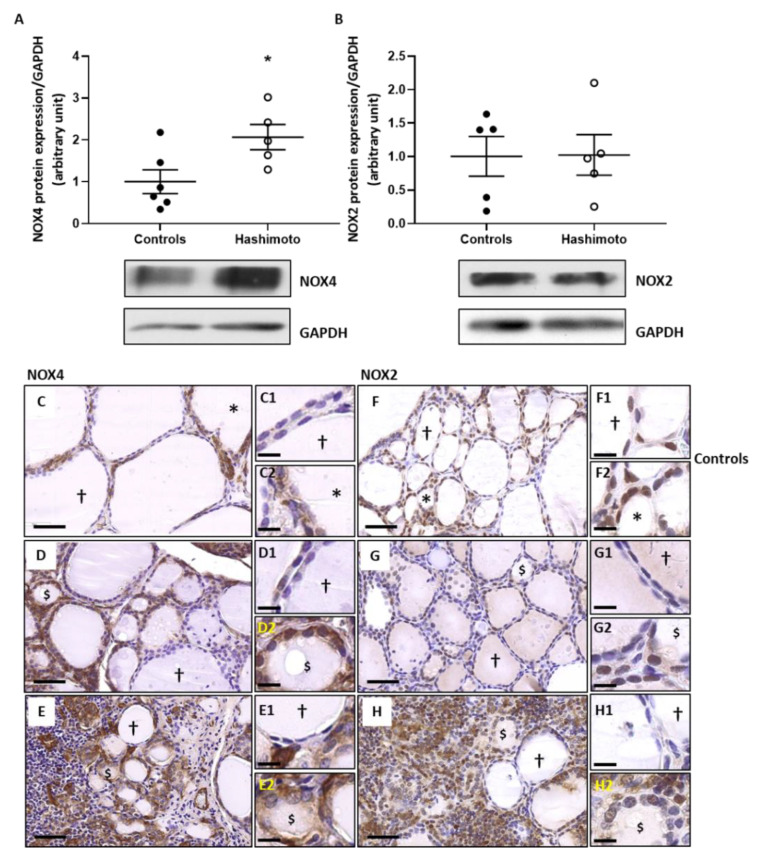
NOX4 protein expression was increased in Hashimoto’s thyroiditis (HT) thyroid samples while NOX2 did not change. Western blots performed on control and Hashimoto’s thyroiditis (HT) thyroid samples revealed a significant increase of NOX4 protein in HT (**A**), but no change in NOX2 (**B**). Densitometric values were normalized against Glyceraldehyde 3-phosphate dehydrogenase (GAPDH) level. Results are expressed as means ± SEM from five or six individual samples (*n* = 5–6). * *p* < 0.05 compared to control. Representative blots are shown. Immunohistochemistry on paranodular tissue obtained from multinodular goiter patients designated as controls (**C**,**F**) and thyroid tissues from HT patients (**D**,**E**,**G,H**). The pictures illustrate representative tissue samples of both conditions. In control thyroids, the expression of NOX4 (**C**) and NOX2 proteins (**F**) was more intense in active follicles (*) than in hypofunctional follicles (†). In HT non-inflammatory zone, hypofunctional follicles (†) lightly expressed NOX4 (**D**) and NOX2 (**G**). In contrast, thyrocytes of altered active-like follicles ($) expressed strongly NOX4 in their cytoplasm (**D**) while NOX2 was moderately expressed (**G**). In HT inflammatory zones, the numerous altered active-like follicles ($) expressed elevated levels of NOX4 in the cytoplasm of their thyrocytes (**E**). NOX2 was highly expressed in inflammatory cells (**H**). (**C**–**H**) Scale bar = 50 µm. (**C1**–**H2**) Scale bar = 20 µm.

**Figure 3 ijms-22-03806-f003:**
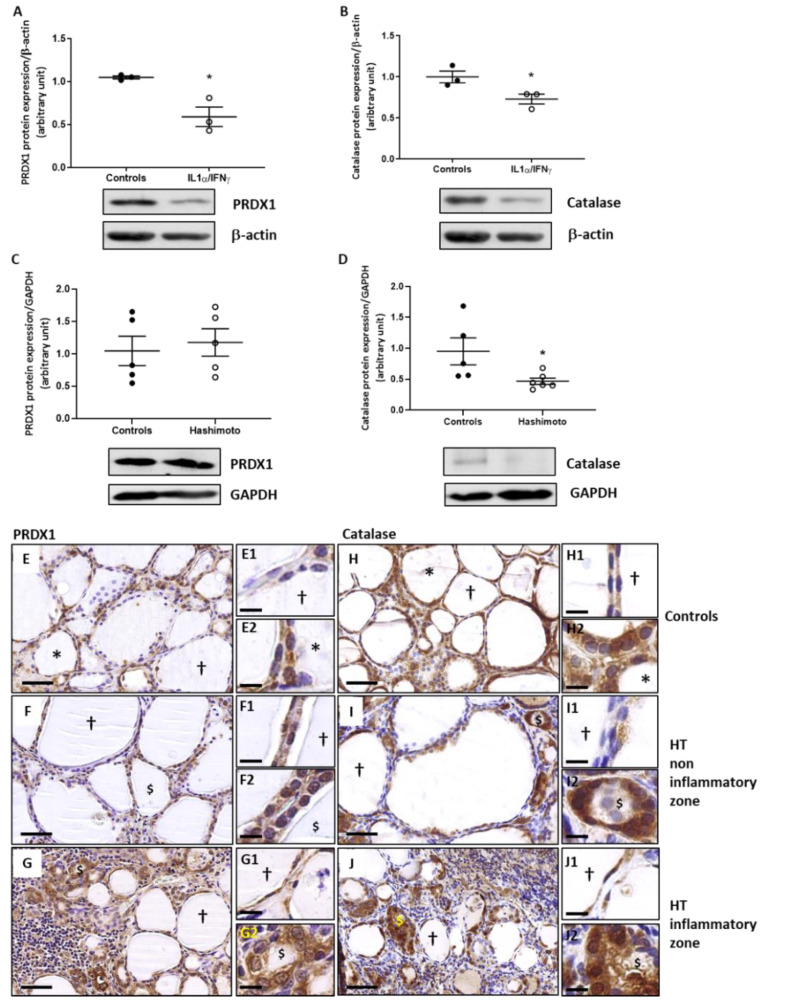
Antioxidant proteins, PRDX1, and catalase, were decreased by Th1 cytokines and catalase was significantly decreased in thyroid samples. PRDX1 and catalase protein expression were quantified by Western blot technique. Primary cultures of human thyrocytes treated with Th1 cytokines had significantly reduced PRDX1 (**A**) and catalase (**B**) expression. Densitometric values were normalized against the β-actin level. Results are expressed as means ± SEM from three experiments (*n* = 3) at least in duplicate. In HT thyroid samples, PRDX1 protein expression (**C**) was unchanged as compared to controls, while catalase protein expression (**D**) was significantly reduced. Densitometric values were normalized against GAPDH level. Values are expressed as means ± SEM from five or six individual samples (*n* = 5–6). * *p* < 0.05, compared to controls. Representative blots are shown. Immunohistochemically stained sections of paranodular tissue from multinodular goiter patients designated as controls (**E**,**H**) and HT thyroid tissues (**F**,**G**,**I,J**). In control glands, hypofunctional follicles (†) expressed low levels of PRDX1 and active follicles (*) showed an increased PRDX1 (E) and catalase (**H**) marking. In HT thyroids, hypofunctional (†) and altered active-like ($) follicles in non-inflammatory zone presented, respectively, no and low staining of PRDX1 (**F**) and of catalase (**I**). In inflammatory zones, active-like follicles ($) strongly expressed PRDX1 (**G**) and catalase (**J**). Illustrated images are representative of both conditions. (**E**–**J**) Scale bar = 50 µm. (**E1**–**J2**) Scale bar = 20 µm.

**Figure 4 ijms-22-03806-f004:**
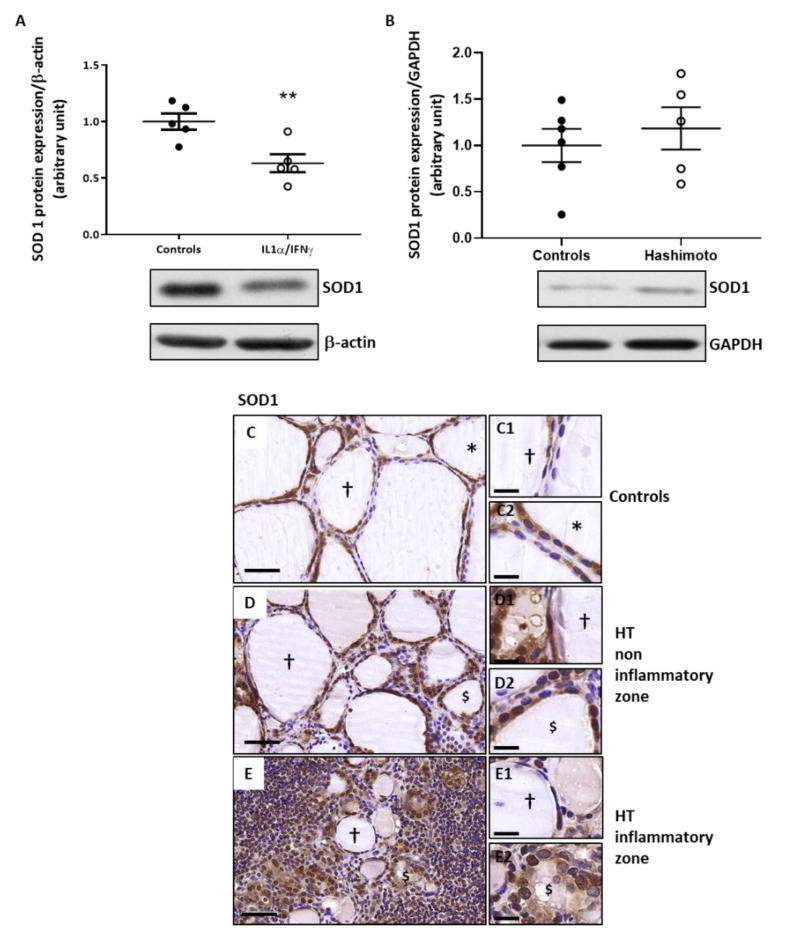
SOD1 protein expression was decreased by Th1 cytokines and unchanged in HT thyroid samples. Human thyrocytes treated with Th1 cytokines showed a significant reduction of SOD1 expression compared to non-treated cells (**A**). Densitometric values were normalized against β-actin level. Results are expressed as means ± SEM from five experiments (*n* = 5) at least in duplicate. ** *p* < 0.01 compared to controls. (**B**) In HT thyroids, SOD1 expression was similar to controls. Densitometric values were normalized against GAPDH level. Values are expressed as means ± SEM from five or six individual samples (*n* = 5–6). Representative blots are shown. SOD1 immunostaining was similar in control (**C**) and HT thyroids (**D**,**E**). Images shown are representative of each condition. (**C**–**E**) Scale bar = 50 µm. (**C1**–**E2**) Scale bar = 20 µm. (†) hypofunctional follicles, ($) altered active-like follicles.

**Figure 5 ijms-22-03806-f005:**
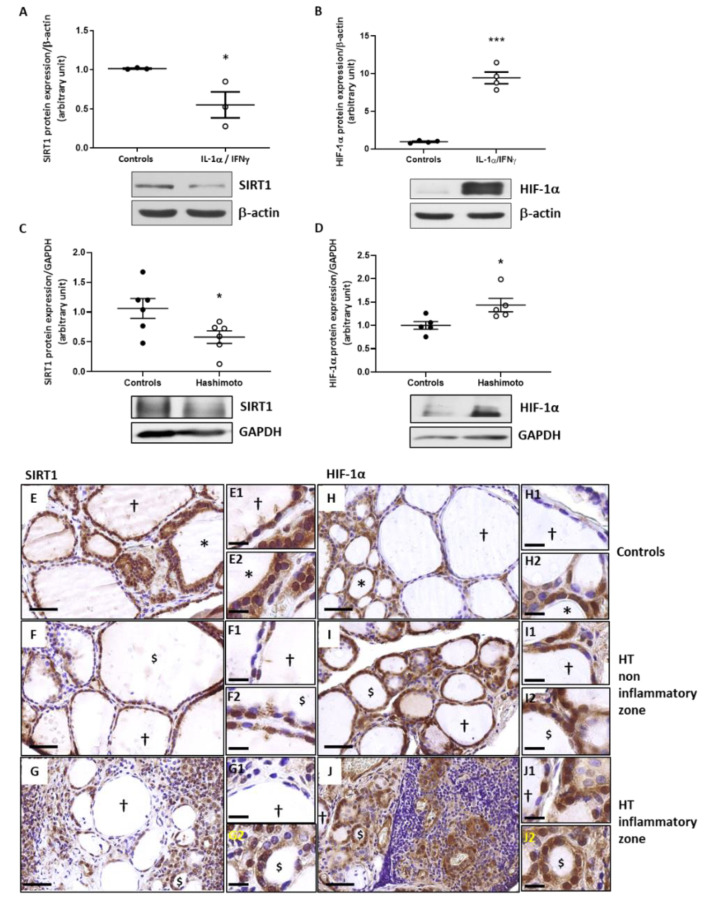
In Th1-HT context, SIRT1 was downregulated while HIF-α was upregulated. Addition of Th1 cytokines to primary cultures of human thyrocytes induced a significant reduction of SIRT1 (**A**) and an increase of HIF-1α protein (**B**). Densitometric values were normalized against the β-actin level. Results are expressed as means ± SEM from seven (SIRT1) and four (HIF-1α) experiments (*n* = 4–7) at least in duplicate. SIRT1 protein expression (**C**) was significantly decreased in HT patients while HIF-1α (**D**) was increased. Densitometric values were normalized against the GAPDH level. Results are expressed as means ± SEM from five or six individual samples (*n* = 5–6). * *p* < 0.05, *** *p* < 0.005 compared to controls. Representative blots are shown. Paranodular tissue from multinodular goiter patients designated as controls (**E**,**H**) and HT thyroid samples (**F**,**G**,**I**,**J**) were used to perform IHC with SIRT1 and HIF-1α specific antibodies. In controls (**E**), SIRT1 was detected in thyrocytes of hypofunctional follicles (†) and more intensively in active follicles (*). In HT (**F**,**G**), staining was less intense in altered active-like follicles ($). In control thyroids (**H**), HIF-1α staining was observed in the cytoplasm of the thyrocytes († and *). In HT thyroids (**I**,**J**), non-inflammatory and inflammatory zones displayed hypofunctional (†) and altered active-like ($) follicles with strong staining of HIF-1α in thyrocyte cytoplasm. Illustrations shown are representative of both conditions. (**E**,**J**) Scale bar = 50 µm. (**E1**–**J2**) Scale bar = 20 µm.

**Figure 6 ijms-22-03806-f006:**
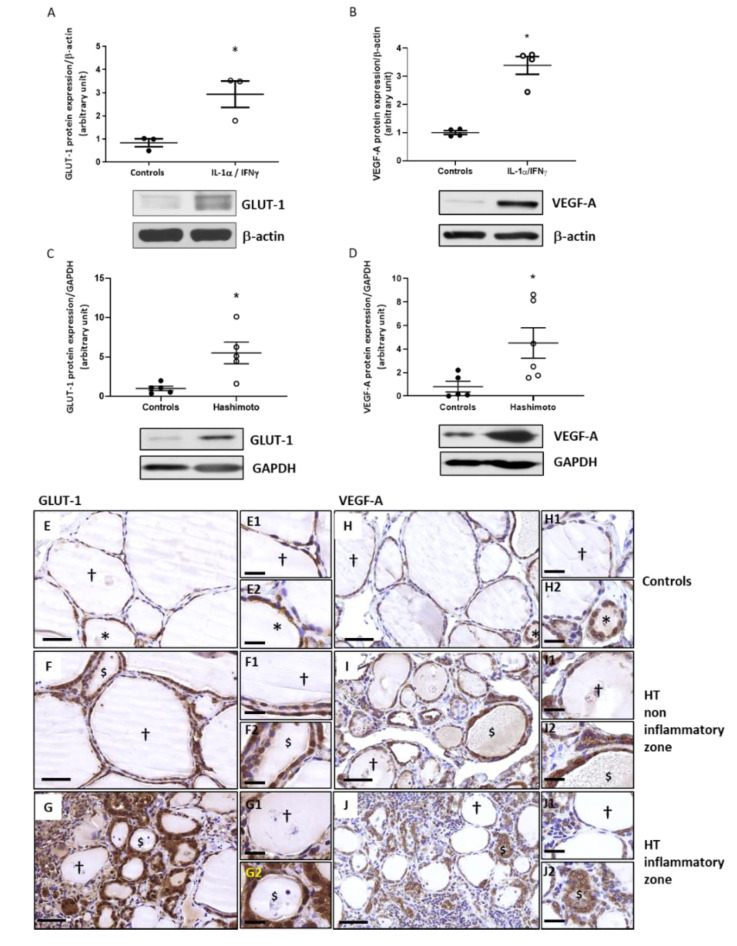
GLUT-1 and VEGF-A expressions were increased in Th1-HT context. GLUT-1 and VEGF-A protein expressions were quantified using Western blot technique. Significantly increased GLUT-1 was observed in thyroid cells treated with Th1 cytokines (**A**). Similar result was obtained for VEGF-A protein (**B**). Densitometric values were normalized against β-actin level. Results are expressed as means ± SEM from three (GLUT-1) and four (VEGF-A) experiments (*n* = 3–4) at least in duplicate. GLUT-1 was highly increased in HT patients compared to controls (**C**). A significant increase of VEGF-A protein expression was also observed in HT thyroids compared to controls (**D**). Densitometric values were normalized against the GAPDH level. Results are expressed as means ± SEM from five or six individual samples (*n* = 5–6). * *p* < 0.05 compared to controls. Representative blots are shown. Faint GLUT-1 immunolabelling was detected in control thyroids (**E**) and the staining was more pronounced in active follicles (*). In non-inflammatory zones of HT thyroids, thyrocytes of altered active-like follicles ($) and hypofunctional follicles (†) showed considerable, diffuse, and cytoplasmic staining of GLUT-1 (**F**). In inflammatory zones, GLUT-1 expression was still more pronounced in active-like follicles ($) (**G**). In control thyroids (**H**), VEGF-A staining was low in hypofunctional follicles (†) and was moderate in active follicles (*). Both hypofunctional (†) and altered active-like follicles ($) in non-inflammatory zone (**I**) and inflammatory zone (**J**) of HT thyroids strongly expressed VEGF-A protein. Images show representative samples. (**E**–**J**) Scale bar = 50 µm. (**E1**–**J2**) Scale bar = 20 µm.

**Figure 7 ijms-22-03806-f007:**
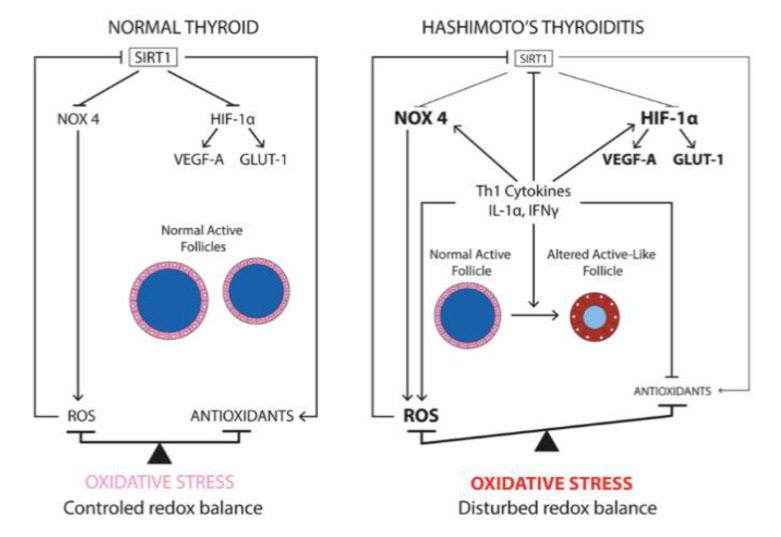
SIRT1 is a new, possible key element in the physiopathogenesis of Th1 autoimmune Hashimoto’s thyroiditis. In normal thyroid, the redox balance is properly controlled and SIRT1 is expressed in the thyrocytes’ cytoplasm, mainly in active follicles. In Hashimoto’s thyroiditis, Th1 cytokines increase ROS production by upregulating NOX4 and decreasing the antioxidant enzymes such as PRDX1, catalase, and SOD1. The disruption of the redox balance induces an oxidative stress, particularly in altered active-like follicles. HIF-1α and its targets, VEGF-A and GLUT-1, are overexpressed by Th1 cytokines and in Hashimoto’s patients. The downregulation of SIRT1 by Th1 cytokines and in Hashimoto’s patients is a key event in the thyroiditis pathogenesis since SIRT1 has been shown in other cell types to be able to modulate oxidative stress by regulating NOX4 and HIF-1α. (T) inhibitor arrow and (↑) activator arrow.

**Table 1 ijms-22-03806-t001:** Characteristics of Hashimoto’s patients.

Number of Hashimoto’s Patients		n = 6
Mean (range) age		46.5 (24–58) years
Sex	Female Male	n = 5 n = 1
Mean (range) HT duration		53.2 (3–156) months
Mean (range) free T4 levels at biopsy time		13.78 (12.1–15.6) pmol/L; n = 5
Mean (range) TSH levels at biopsy time		2.39 (0.01–4.22) µU/mL: n = 6
Mean (range) anti-TPO levels at biopsy time		232.675 (67–491.8) U/l; n = 4
L-thyroxine treatment		n = 6
Medical imagery		Suspicious nodules; n = 4

**Table 2 ijms-22-03806-t002:** Experimental conditions for immunohistochemistry (IHC) and Western blot (WB).

Proteins	Primary Antibodies	References	Incubation Conditions	
			IHC	WB
Catalase	Mouse Monoclonal Antibody (Sigma, Merck KGaA, Darmstadt, Germany)	C-0979	1 µg/mL, 1 h	2 µg/mL, overnight
GAPDH	Rabbit Monoclonal Antibody (Cell signaling, Leiden, The Netherlands)	2118S	/	1/3000, overnight
GLUT-1	Rabbit Polyclonal Antibody (Abcam, Cambridge, UK)	Ab32551	1/200, overnight	1/1000, overnight
HIF-1α	Rabbit Polyclonal Antibody (Novus Bio, Oxon, UK)	NB 100449	4 µg/mL, overnight	0.4 µg/mL, overnight
HNE	Rabbit Polyclonal Antibody (Sigma, Merck KGaA, Darmstadt, Germany)	393207-100	1/200, overnight	/
NOX2	Rabbit Polyclonal Antibody (Abcam, Cambridge, UK)	Ab31092	5 µg/mL, overnight	0.5 µg/mL, overnight
NOX4	Rabbit Polyclonal Antibody (Proteintech, Manchester, UK)	14347-1-AP	1/200, overnight	1/1000, overnight
PRDX1	Rabbit Polyclonal Antibody (Abcam, Cambridge, UK)	Ab15571	1/100, overnight	1/1000, overnight
SIRT1	Rabbit Monoclonal Antibody (Cell signaling, Leiden, The Netherlands)	9475S	/	1/1000, overnight
SIRT1	Mouse Monoclonal Antibody (ThermoFischer, Merelbeke, Belgium)	MA5-15677	1/150, overnight	/
SOD1	Rabbit Polyclonal Antibody (Abcam, Cambridge, UK)	Ab13498	1/500, overnight	1/1000, overnight
T4	Mouse Monoclonal Antibody (MyBiosource, San Diego, CA, USA)	MBS 5920 059	1/400, overnight	/
VEGF-A	Rabbit Monoclonal Antibody (Abcam, Cambridge, UK)	Ab46154	5 µg/mL, overnight	0.5 µg/mL, overnight
β-actin	Mouse Monoclonal Antibody (Sigma, Merck KGaA, Darmstadt, Germany)	A5441	/	1/200,000, overnight

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
