# Peer review of "Oxidative Stress-Induced Sirtuin1 Downregulation Correlates to HIF-1α, GLUT-1, and VEGF-A Upregulation in Th1 Autoimmune Hashimoto’s Thyroiditis"

_ijms, 2021, doi:10.3390/ijms22083806_

Round 1
Reviewer 1 Report
This is interesting report.
Author Response
Dear Reviewer,
We would like to thank you for having accepted to review our paper entitled “Oxidative stress-induced Sirtuin1 downregulation correlates to HIF-1α, GLUT-1 and VEGF-A upregulation in Th1 autoimmune Hashimoto's thyroiditis”. As indicated afterwards, we took into account all issues and adapted the manuscript accordingly.
This is interesting report.
As requested, the manuscript was revised for English language and style by Professor Marian Ludgate, Professor Emerita of Molecular Endocrinology, Cardiff University.
Reviewer 2 Report
The aim of this study is to investigate SIRT1 as a key player and therapeutic target in Hashimoto’s thyroiditis. In fact, SIRT1 was downregulated in Hashimoto's thyroiditis patient compared to controls. Although this manuscript is interested, this is not enough and deeply for novel research as regular paper. I propose that to improve and to increase potential impact figures in this field of scientific research.
Major comments
Comment 1
In this study, author directly indicated that H2O2 decreased catalase in Hashimoto's thyroiditis patient compared to control (Fig.3). As well as Fig. 4, SOD expression level in Hashimoto's thyroiditis patient did not change compared to control. I can’t understand and see between indicted data and manuscript. Author should submit figures that match the statistics as much as possible. Or author should be show multiple samples at once.
Comment 2
Do you have a mimic in a cell culture system about Hashimoto's thyroiditis? Author should measure another chemokine such as CXCL10, CCL2 and CXCL19.
Comment3
Several researchers reported that SIRT1 was downregulated by H2O2 treatment in various cell lines. In this study, oxidative stress induced sitr1 downregulation leased to HIF-GLUT and VEGF-A upregulation in Hashimoto’s thyroiditis. This statement is too drastic. As same as discussion, I ask you to change it because it is too dramatics. I'll request an additional experiment such as IL-1/IFNB-gamma with NAC treatment etc. Because, it is unclear whether downregulation of SIRT1 by oxidative stress actually increases HIF-1, Glut4 and VEGF-A expression. This mechanism is very important.
Author Response
Dear Reviewer,
We would like to thank you for having accepted to review our paper entitled “Oxidative stress-induced Sirtuin1 downregulation correlates to HIF-1α, GLUT-1 and VEGF-A upregulation in Th1 autoimmune Hashimoto's thyroiditis”. As indicated afterwards, we took into account all issues and adapted the manuscript accordingly.
The aim of this study is to investigate SIRT1 as a key player and therapeutic target in Hashimoto’s thyroiditis. In fact, SIRT1 was downregulated in Hashimoto's thyroiditis patient compared to controls. Although this manuscript is interested, this is not enough and deeply for novel research as regular paper. I propose that to improve and to increase potential impact figures in this field of scientific research.
As requested, the manuscript was revised for English language and style by Professor Marian Ludgate, Professor Emerita of Molecular Endocrinology, Cardiff University.
Major comments
Comment 1
In this study, author directly indicated that H2O2 decreased catalase in Hashimoto's thyroiditis patient compared to control (Fig.3). As well as Fig. 4, SOD expression level in Hashimoto's thyroiditis patient did not change compared to control. I can’t understand and see between indicated data and manuscript. Author should submit figures that match the statistics as much as possible. Or author should be show multiple samples at once.
We would like to precise that we have not directly indicated that H2O2 decreased catalase in Hashimoto’s thyroiditis (HT) patients compared to controls. Figure 3 shows a catalase reduction by Th1 cytokines or in HT samples and the role of H2O2 has only been suggested in the discussion.
Indeed, SOD1 expression was not significantly decreased in HT samples but this must be related to the heterogeneity among patients and within a thyroid sample. This follicular heterogeneity is a hallmark of the normal thyroid gland and it is increased with ageing and in various pathological conditions. This has been indicated in the discussion.
As far as we know, all the figures of the manuscript match the statistics.
Powerpoint file with all the blots that have been analysed has been included with the first submission.
Comment 2
Do you have a mimic in a cell culture system about Hashimoto's thyroiditis? Author should measure another chemokine such as CXCL10, CCL2 and CXCL19.
The role of IL-1a and IFNg in HT is well known. Both cytokines are classically used on thyroid cell lines and human thyroid cells in primary cultures showing that they alter thyrocytes function inducing ROS production and thereby oxidative stress (Sato et al, 1990 ; Ajjan et al, 1998 ; Rasmussen et al, 2000; Caraccio et al, 2005 ; Gerard et al, 2006 ; Poncin et al, 2008)
Indeed, chemokines like CXCL10, CCL2 and CXCL19 are known to be involved in autoimmune thyroid disorders and produced by thyrocytes in response to IFNg (Antonelli et al, 2014) but they could not be considered as specific for HT since they are also produced in Graves’ disease.
Comment 3
Several researchers reported that SIRT1 was downregulated by H2O2 treatment in various cell lines. In this study, oxidative stress induced sitr1 downregulation leased to HIF-GLUT and VEGF-A upregulation in Hashimoto’s thyroiditis. This statement is too drastic. As same as discussion, I ask you to change it because it is too dramatics.
We agree with the comment and have modified the title and the discussion to be less drastic.
I'll request an additional experiment such as IL-1/IFNB-gamma with NAC treatment etc. Because, it is unclear whether downregulation of SIRT1 by oxidative stress actually increases HIF-1, Glut4 and VEGF-A expression. This mechanism is very important.
It’s a very interesting suggestion. However, the requested additional experiment using NAC treatment has already been performed by our group showing that NAC reduces ROS production induced by IL1a and IFNg (Poncin et al, 2010 ; Colin et al, 2014).
Reviewer 3 Report
International Journal of Molecular Biosciences
Manuscript No.: ijms-1104928-peer-review-v1
Title: Oxidative stress-induced Sirtuin1 downregulation leads to HIF-2α, GLUT-1 and VEGF-A upregulation in Th1 autoimmune Hashimoto's thyroiditis
Authors: Michaël Hepp, Alexis Werion, Axel De Greef, Christine de Ville de Goyet, Marc de Bournonville, Catherine Behets, Benoit Lengelé, Chantal Daumerie, Michel Mou-rad, Marie-Christine Many, Virginie Joris and Julie Craps
This is an interesting study examining the thyroid autoimmune disorder Hashimoto's thyroiditis (HT) and the role of oxidative stress in the pathophysiology of the disease. The authors determine the involvement of NADPH-oxidases (NOX), sirtuin 1 (SIRT1) and the hypoxia-inducible factor-1 alpha (HIF-1α) in the HT pathophysiology as well as the role of antioxidant proteins such as peroxiredoxin (PRDX1), catalase and superoxide dismutase (SOD1). The authors use human thyroid cells that were exposed to Th1 cytokines (IL-1α and IFNγ) and HT human thyroid tissue samples as their model systems. This study follows up on a paper that the group published in 2014 on the expression of dual oxidase, thyroid peroxidase and caveolin-1 (Marique et. al. J. Clin. Endocrinol. Metab 2014, 99, 1722-1732; Reference 8) and a paper published in 2015 which examined the expression of PPARg, caveolin-1 and catalase (Werion et. al. Thyroid 2016, 26, 1320-1331; Reference 9) in HT patients. These studies also examined markers of oxidative stress (OS), cell proliferation, apoptosis, and antioxidant defenses.
The manuscript, in general, is well written, but would benefit from clarification of the results presented. This reviewer recommends major revisions for this manuscript to be published in IJMS. In revising the manuscript for publication in IJMS, this reviewer hopes that the authors take the following points into consideration.
Major
- In immunohistochemical samples, are hypofunctional and active follicles differentiated by their small size and their thick thyrocytes alone? There may be intermediates in between these two states and some of these can be seen in the figures presented. Are there other immunohistochemical markers that could be used to differentiate between these two follicle types?
- Figure 2: While the authors show an increase in NOX4, but not NOX2, in HT thyroid samples in this figure, and suggest that this results in higher oxidative stress (OS), they do not use any measure of OS in this experiment. Measurements of HNE, as suggested by the Marique et al (2014) reference, could have been carried out on these samples to show that they are, indeed, undergoing OS.
- Figure 2: It is also unclear in this figure how NOX4 and NOX2 are stained as there seems to be little difference between panels C versus F, D versus G and E versus H. The authors state that “the expression of NOX4 and NOX2 proteins is more intense in active-like follicles (*) than in hypofunctional follicles (†)” but to this reviewer the staining seem to be indistinguishable between these two compartments. The Western blots on the protein expressions of NOX4 and NOX2 are clear, but the immunostaining is not.
- Figure 3D: While the densitometry shows a decrease in catalase protein, the representative Western blot used for this figure shows little change in this protein.
- Figure 3: Like Major point 3, it is difficult to see staining differences between PRXN1 and catalase staining patterns between hypofunctional (†) and altered active-like (*) follicles in non-inflammatory zones of HT patient thyroids. Similar issues are found in the immunohistochemistry panels of Figures 4 (SOD1), 5 (SIRT1 and HIF-1α), and 6 (GLUT1 and VEGF-A). It is unclear to this reviewer differences in staining between hypofunctional and altered active-like follicles in these figures.
- Thyrocytes treated with IL-1α and IFNγ showed a highly increased HIF-1α protein expression compared to non-treated cells. It was not clear why HIF-1α was examined at all in this study? The cells were not treated with either hypoxia (Reference 14) or iodide deprivation (References 15 and 16), and it is unclear how HIF-1α was upregulated in this system. If it is through the Th1 cytokines used in this study alone, there is some question as to why IL-1α and IFNγ were used. Previous studies have shown that it is IL-1β, and not IL-1α, that upregulates HIF-1α in various cellular systems. The authors should comment on this.
- The use of CM-H2DCFDA to measure reactive oxygen species (ROS) is problematic as a) it has been found to autofluoresce and b) has been found to leak out of cells once it is deacetylated by intracellular deacetylases. Did the authors attempt to use other, more reliable, fluorescent dyes to measure ROS?
- There are many similarities and overlaps with the author’s previous two papers in 2014 and 2016. This includes the measurement of catalase.
- The statement “Our data support the hypothesis that SIRT1 acts as an inhibitor of HIF-1α since we observed lower levels of SIRT1 associated with a higher HIF-1α protein expression in Th1-treated human thyroid cells and in HT thyroid samples as compared to control” is an overstatement. The effects shown by the authors are correlative. The authors did not manipulate the systems (overexpression or knock down/knock out) to show a direct causative effect. The authors also state that other factors, such as “post-translational modifications, inflammatory cytokines and miRNA” could also affect SIRT1 and HIF-1a function, and these could be at play in their system.
- Samples from Hashimoto's patients were taken from 5 female and 1 male patient. Were these samples mixed for the present study? Are there sex differences that may play a role in the responses to Th1 cytokines in these patients?
Minor
- Lines 27-32: Change “Sirtuin (SIRT)1” to “Sirtuin 1 (SIRT1)”. Change “peroxiredoxin (PRDX)1” to “peroxiredoxin 1 (PRDX1)”. Change “superoxide dismutase (SOD)1” to “superoxide dismutase 1 (SOD1)”. Change “Glucose Transporter (GLUT)-1” to “Glucose Transporter 1 (GLUT1)”. Change “Vascular Endothelial Growth Factor (VEGF)-A” to “Vascular Endothelial Growth Factor A (VEGF-A)”.
- Line 62: Change “transforming O2.- in H2O2[7]” to “transforming superoxide into H2O2 [7]”. Likewise change “” to “superoxide” throughout the manuscript as the “dot is difficult to see and should be superscripted. Spaces should be inserted between the citations “[X]” and the word before it throughout the manuscript.
- Line 90: Change “ratio and thus the intracellular redox status seems” to “ratio, and thus the intracellular redox status, seems”.
- Line 92: Change “Up to date” to “To date”.
- Lines 100-103: Change “We analyzed 1° the expression of NOX2 and NOX4 as sources of ROS, 2° the expression of antioxidants: PRDX1, catalase and SOD1, 3° the expression of SIRT1 protein in the context of the HT oxidative stress, 4° the expression of HIF-1α and its targets: GLUT-1 and VEGF-A” to “We analyzed 1) the expression of NOX2 and NOX4 as sources of ROS, 2) the expression of antioxidants: PRDX1, catalase and SOD1, 3) the expression of SIRT1 protein in the context of the HT oxidative stress and 4) the expression of HIF-1α and its targets, GLUT-1 and VEGF-A”.
- Lines 126, 158, 185, 188, 210, 212-213, 234, 236-237, 264: Change “mean ±SEM” to “means ± SEM”.
- Lines 134: Change “The whole thyroid samples” to “Whole thyroid samples”.
- Line 144: Define HNE first usage.
- Line 183: Change “technic” to “technique”.
- Line 283: Change “trend” to “trends”.
- Line 284: Change “active follicles the latter being” to “active follicles; the latter being”.
- Lines 298-299: Single line paragraph. Should be removed and combined with another paragraph.
- Line 489: subscript the “2” in superoxide or remove the acronym altogether and just simply use “superoxide”.
- Table 1: Line numbers overlap with the table itself, making parts of it hard to read.
- References: Some article titles have all words capitalized and others have only the first word capitalized. The authors should double check the formatting and adhere to the reference format in the IJMS Guide To Authors.
Author Response
Please you can find the point-by-point response to your comments in the attached word file.

Reviewer 4 Report
Review of the manuscript entitled
‘Oxidative stress-induced Sirtuin1 downregulation leads to HIF-1α, GLUT-1 and VEGF-A upregulation in Th1 autoimmune Hashimoto's thyroiditis’
by Michaël Hepp, Alexis Werion, Axel De Greef, Christine de Ville de Goyet, Marc de Bournonville, Catherine Behets, Benoit Lengelé, Chantal Daumerie, Michel Mourad, Marie-Christine Many, Virginie Joris and Julie Craps
The authors in the presented manuscript entitled ‘Oxidative stress-induced Sirtuin1 downregulation leads to HIF-1α, GLUT-1 and VEGF-A upregulation in Th1 autoimmune Hashimoto's thyroiditis’ confirm the role of oxidative stress (OS) in the pathogenesis of Hashimoto's thyroiditis (HT) and indicate SIRT1 as a regulator of redox imbalance in thyrocytes. Furthermore, they propose a mechanistic explanation including SIRT1/NOX4 and SIRT1/HIF-1α/GLUT-1 and VEGF-A axes.
The pathogenesis of HT is complex and not fully understood. The role of OS in the pathogenesis of HT is more frequently discussed. The presented manuscript nicely fits into this trend.
In my opinion, the work is carefully prepared and thus deserves to be published after minor corrections. Below I enclose a list of some issues that should be improved before publishing.
- To make the Western blot data more reliable the protein expression from tissue samples and primary cultured cell lines should be normalized to the same reference protein.
- Please include the information how many repeats have been conducted in one experiment.
- The manuscript requires minor language correction including some unfortunate wording such as “Th1 cytokines added on human primary thyrocytes induce a significant reduction of SIRT1” (line 232).

Author Response
Please you can find in attached word file the point-by-point answer to your comments.

Reviewer 5 Report
This study combines the use of primary cells and human tissues to assess alterations to antioxidant proteins (primarily those involved in the superoxide radical anion - hydrogen peroxide axis), HIF-alpha and downstream proteins Glut 4/VEGF and explored a link to these changes with alteration in Sirtuin protein expression in the same tissues. First I complement the authors on the extensive assessment of protein regulation as this is superior to approaches that assess gene regulation alone. The approach to reviewing statistcial differences is also complemented and many other researchers should adopt the approach to test for normalcy first and adjust the statistiial test based on the outcome determined.
Several issues have been identified on review of the submitted manuscript as follows:
1. One major limitation of the study is that the tissue is showing regional differences in the extent of inflammation/protein regulation. Thus, when the Western blotting is done the tissue is homogenised and the regional differences are no longer able to be considered. If possible dissecting tissue on the basis of inflammatory status wold go to clarifying the actual changes protein levels for the antioxidant protein, Glut, HIF1alpha and Sirtuin proteins in the different regions of the tissue being assessed. Notably, the data shown indicates that localised / regional changes in these protein expression may be important in host tissue responses. IF the authors were to adopt this approach then it would be easier to justify the schematic shown in Figure 7.
2. The data presented is visually interesting but this mode provides is largely observational assessments. Western blotting is also semi-quantitative. It would be instructive for the authors to include some biochemical assessments of protein activity to match the regulation that is being indicated by the visual modes of presentation. Thus NOS activity assays and antioxidant activity assays are avaialble for a majority of the proteins being assessed and this complementary data would be confirmatory and add to the robust presentation of the data.
3. The data shown in IHC is reasonably convincing for total change. For example, levels of NOX4 are elevated in the tissue but what cell types are involved? This is important to understand as well as the time course for the cellular activation. Improved microscopy (higher level of magnification) is required to demonstrate this better. Overall, it is difficult to distinguish the contribution of endogenous cells from cells recruited to the inflammed regions. The same can be said of the other proteins analysed using this same IHC approach.
4. The observation that enzymes involved in consumption of H2O2 are selectively regulated is very interesting. Also, PRDX1 can be over-oxidised and alter activity. As stated above, the authors should consider adding activity data to corroborate these immuno-histological assessments; this may have to involve sectioning the tissue to reflect non-inflammed and inflammed regions to correlate the altered protein expression/ corresponding activity with the tissue response status.
5. This homogenising method to obtain material for Western blotting studies removed the possibility to review regional differences in the tissue and effectively averaged protein levels across differentially affected tissues - this is a major limitation of this study and needs to be highlighted in the discussion.
6. The CM-H2DCFDA probe can autocatalyze its own oxidation as it also produces H2O2. Thus, when using this probe it is best to use a positive control to remove H2O2 to demonstrate the probe is measuring a ROS within the tissue and not generated from itself and/or use complimentary approaches to identify ROS damage in the same tissues in the absence of the probe.
Author Response
Please you can find in attached word file the point-by-point answers to your comments.

Round 2
Reviewer 2 Report
Author's manuscript has been improved and answers our questions clearly. We accept this manuscript.
Reviewer 5 Report
The auhtors ahve addressed most fo the major points and have added some additional data to complement the single assessment of ROS that was a concern/limitation in the previous version.
Where they have not been able to address concerns experimentally the authors have made an effort to outline limitations of their study.
I have no further issues with the revised manuscript.